# Identification of Appropriate Endogenous Controls for Circulating miRNA Quantification in Working Dogs under Physiological Stress Conditions

**DOI:** 10.3390/ani13040576

**Published:** 2023-02-06

**Authors:** Gabriella Guelfi, Camilla Capaccia, Michele Matteo Santoro, Silvana Diverio

**Affiliations:** 1Laboratory of Ethology and Animal Welfare (LEBA), Department of Veterinary Medicine, Università degli Studi di Perugia, Via San Costanzo 4, 06126 Perugia, Italy; 2Department of Veterinary Medicine, Università degli Studi di Perugia, Via San Costanzo 4, 06126 Perugia, Italy; 3GdF (Military Force of Guardia di Finanza), Dog Breeding and Training Course, Via Lungolago 46, 06061 Castiglione del Lago, Italy

**Keywords:** endogenous control, qPCR, circulating miRNA, search and rescue dogs, physiological stress

## Abstract

**Simple Summary:**

Circulating miRNAs are not only present in cells but also in the extracellular environment, especially in different biofluids including blood, and can act in a paracrine manner by facilitating a diversity of signaling mechanisms between cells. In qPCR gene expression profiling analysis, the endogenous control must be a stable gene to allow an accurate cross-sample gene expression comparison. The appropriate selection of an endogenous control is a crucial step. This research aims to select, in the miRNome, appropriate circulating miRNAs that can serve as an endogenous control. The study model are working dogs used in the search and rescue of missing persons after natural disasters.

**Abstract:**

Cell-free miRNAs, called circulating miRNAs (cmiRNAs), can act in a paracrine manner by facilitating a diversity of signaling mechanisms between cells. Real-time qPCR is the most accepted method for quantifying miRNA expression levels. The use of stable miRNA endogenous control (EC) for qPCR data normalization allows an accurate cross-sample gene expression comparison. The appropriate selection of EC is a crucial step because qPCR data can change drastically when normalization is performed using an unstable versus a stable EC. To find EC cmiRNA with stable expression in search and rescue (SAR) working dogs, we explored the serum miRNome by Next-Generation Sequencing (NGS) at T0 (resting state) and T1 immediately after SAR performance (state of physiologically recovered stress). The cmiRNAs selected in the NGS circulating miRNome as probable ECs were validated by qPCR, and miRNA stability was evaluated using the Delta Ct, BestKeeper, NormFinder, and GeNorm algorithms. Finally, RefFinder was used to rank the stability orders at both T0 and T1 by establishing miR-320 and miR-191 as the best-circulating ECs. We are confident that this study not only provides a helpful result in itself but also an experimental design for selecting the best endogenous controls to normalize gene expression for genes beyond circulating miRNAs.

## 1. Introduction

MicroRNAs (miRNAs) are small non-coding RNAs that regulate target protein expression through two primary mechanisms: translational blocking in the initiation step or the elongation phase and removal of the polyA tail, promoting deadenylase activity followed by mRNA degradation [1,2,3]. Recently, it has been found that miRNAs are not only present in cells but also in the extracellular environment, particularly in various biofluids [4,5]. Cell-free miRNAs, called circulating miRNAs, can act in a paracrine way by facilitating a diversity of signaling mechanisms between cells [6]. In comparison to intracellular miRNAs, cmiRNAs are extremely stable thanks to a kind of packaging that is crucial to protect the miRNA from digestion by RNases present in body fluids [7]. Roughly 10% of cmiRNAs are secreted in microparticles such as exosomes, microvesicles, and apoptotic bodies, while the remaining 90% form complexes with RNA ligands such as lipoprotein complexes such as high-density lipoprotein (HDL), Argonaute2, and nucleophosmin I [5,8]. CmiRNAs play an important role as noninvasive diagnostic biomarkers, and aberrant expression of cmiRNAs is associated with homeostatic imbalances [9] or disease states.

Real-time qPCR is the most accepted method for quantifying miRNA expression levels, thanks to its high sensitivity, reproducibility, specificity, and simplicity of execution. The qPCR results are expressed as a quantification cycle (Cq), and to analyze the changes in gene expression, the Cq is transformed into a relative value using the 2^−ΔΔCq (Livak-Schmittgen) approach. The 2^−ΔΔCq considers, at the exponent, two Δs: the first Δ represents the normalized expression value of the target gene (2^−ΔCq) compared to a stable endogenous control (EC), while the second Δ represents the relative expression (2^−ΔΔCq) of the normalized value versus the experimental control sample. The use of stable miRNA ECs for qPCR data normalization allows an accurate cross-sample gene expression comparison. The appropriate selection of EC is a crucial step to minimize the influence of RNA quality, reverse transcription efficiency, and PCR reaction conditions on the data [10]. qPCR data can change drastically when the data normalization is performed using an unstable versus a stable reference EC. The basic assumption of reference genes as ECs is that they should be characterized by stable expression in each of the samples tested, despite the impact of experimental factors, disease stage, or treatments [11,12].

To date, no prior comparative analysis of eligible EC serum miRNAs in search and rescue (SAR) working dogs in response to physiological conditions of stress and rest has been published [13]. SAR dogs are a valuable asset for locating missing persons after natural disasters. SAR dogs must be in optimum physical condition and have good mental focus, as SAR operations cause severe physical and emotional fatigue in dogs. The fundamental idea of this research is to identify cmiRNA ECs that meet the EC criteria, have a stable expression in the baseline state state (T0) and immediately post-SAR (T1). Where T1 indicates a state of physiologically recovered stress. To achieve this goal, the serum miRNome was explored by Next-Generation Sequencing to identify a panel of miRNAs not differentially expressed at T0 and T1. The miRNAs selected in the NGS circulating miRNome as probable ECs were validated by qPCR, and miRNAs’ stability was evaluated using, the most widely cited gene expression analysis algorithms in the literature, including the Delta Ct method, BestKeeper, NormFinder, and GeNorm. Finally, a comprehensive tool named RefFinder was used to rank the stability order of the selected reference genes.

This study aims to achieve two important objectives: (1) to provide suitable circulating EC miRNAs to study working dogs under resting conditions and SAR physiological stress; and (2) to provide an appropriate method to select the most stable ECs to normalize qPCR data for future gene expression studies.

## 2. Materials and Methods

The research protocol was authorized by the Ethics Committee of the University of Perugia, protocol no. 2018-21 and conforms to the laws of the Italian Ministry of Health. There is a standing agreement between the Department of Veterinary Medicine of the University of Perugia and GdF (Guardia di Finanza) for the ethical investigation and testing of GdF working dogs. All methods were carried out following sound guidelines and regulations, and the study was carried out in compliance with the ARRIVE guidelines. Informed consent is not required as no human subjects were included in the study.

### 2.1. Animal Enrolment

The dogs included in this research work are all experts in avalanche or rubble search because they were trained at the GdF Dog Breeding and Training Center [14] (Castiglione del Lago, Perugia, Italy) and the SAR Alpine School (Passo Rolle, Trento, Italy). The dogs were physically and behaviorally tested [14] to certify their suitability for SAR training and work. Table 1 shows the dogs enrolled in the study.

### 2.2. Experimental Flowchart

SAR trial. The simulated SAR trial required dogs to find a target odor (a hidden operator and his breath, simulating a missing person) on a rubble field (30 × 35 m), within a maximum time of 15 min. Blood was collected at T0 immediately before the SAR (resting baseline) and at T1 later in the SAR (physiological stress). T0 versus T1 enables a relation between two different physiological states, where T1 comprises a multifactorial combination of stresses physiologically recovered in trained dogs [15].

The experimental framework of the study is divided into two phases.In the first step, dog blood was drawn at T0 (n = 22) and T1 (n = 22) (dogs reported in Table 1), and total serum RNA was isolated. The total RNA of six dogs listed in Table 1 (dog ID: A, B, C, D, E, and F) was used for library preparation and NGS miRNome sequencing. This step aimed to evaluate the T0 versus T1 unregulated circulating miRNAs, i.e., candidate stable EC miRNAs.In the second step, unregulated cmiRNAs selected as candidate EC were validated by qPCR in a larger panel of dogs, including all dogs listed in Table 1 (T0 n = 22; T1 n = 22). Then, bioinformatic analysis of the qPCR results identified the EC miRNA stability values and the EC final ranking (Figure 1).

### 2.3. First Step: T0 and T1 Blood Sampling and Serum RNA Extraction

At T0 and T1, 3 mL of the blood is collected as described by Guelfi et al. [15]. Hemolysis was controlled in all serum samples because a significant source of variation in serum comes from contamination by cellular-derived miRNAs resulting from hemolysis. The serum was first visually inspected, and the presence of a pink color indicated the presence of free hemoglobin. Later, hemoglobin concentrations are evaluated in all processed serum samples by the optical density at 414 nm (absorbance peak of free hemoglobin) using a NanoDrop™ 1000 spectrophotometer (Thermo Scientific, Scoresby, Victoria, Australia). If the absorbance of oxyhemoglobin at 414 nm exceeded a value of 0.2, the samples were classified as hemolyzed and were excluded from the experiment. Total RNA was extracted from 200 µL of serum, and quantitative analysis of miRNAs was performed as described in Guelfi et al. [15]. The RNA was divided into two aliquots, one for the first step and another for the second step, and both aliquots were stored at −80 °C until use. To survey RNA extraction, cDNA production, and qPCR, control quality (QC) spike-ins were used. During RNA extraction, UniSp2 and UniSp4 spike-ins were added to assess the efficiency and yield of the RNA isolation, while UniSp6 was included in the RT to observe cDNA synthesis efficiency and control PCR inhibitor presence [15]. In qPCR, UniSp2, UniSp4, and UniSp6 spike-ins were amplified and quantified with the respective primer pair.

### 2.4. First Step: NGS Circulating miRNome and Stable miRNAs Selection

T0 and T1 total RNA (5 µL) were converted into miRNA NGS libraries. Library preparation and next-generation sequencing to select candidate EC miRNAs were performed as described in Guelfi et al. [15]. NGS output data were utilized to identify the most highly expressed and most stable miRNAs in the T0 and T1 miRNomes. The comparison of the T0 and T1 miRNome allowed us to screen, within a broad physiological state, the most stable miRNAs. As a preliminary approach, NGS miRNAs are ordered by the count number to choose only those miRNAs abundantly expressed in the miRNome, according to the dogma that EC miRNAs should be highly abundant in cells and tissues [16]. NGS data are shown in the Appendix A. Then, only the most stable miRNAs, i.e., with T0 versus T1 *p* > 0.05 significance value, FC value < 1.1, and Bonferroni test value > 1, were selected. In the panel of candidate miRNAs for EC, cfa-miR-320 was included because, in previous research, it was elected as the best EC miRNA in dog serum [15]. NGS data are shown in the Appendix A.

### 2.5. Second Step: Probable EC miRNA qPCR Validation

MiRNAs selected as probable ECs were validated by qPCR in an aliquot of serum RNA, extracted in the first step, at T0 (n = 22) and T1 (n = 22). Total RNA (10 ng) was reverse transcribed as described in Guelfi et al. [15]. QPCR amplification was carried out using 3 μL of cDNA (diluted 1:50) and appropriate primers as reported in Guelfi et al. [10] (Table 2).

PCR cycling conditions included initial denaturation for 15 s at 95 °C; followed by 40 cycles of amplification: 15 s at 94 °C for denaturing double-stranded DNA, 15 s at 95 °C for annealing, and 15 s at 70 °C extension steps. Data acquisition should be performed during the annealing/extension step. Melting curve analysis at 60–95 °C was performed to assess amplification specificity. The amplification and the results of interpreting spike-ins were performed, referring to Guelfi et al. [15]. Only reactions with efficiencies ranging from 95 to 100% were included in the subsequent analysis.

### 2.6. Second Step: Bioinformatics Analysis of EC Expression Stability and EC Final Ranking

Subsequently the Cq of validated cfa-miRNAs (320, 486, 16, 92a, 191, 223, 423a, le-7b, 25, and 93) were examined with bioinformatics algorithms, including the comparative ΔCt method [17], BestKeeper [18], NormFinder [19], and GeNorm [20], to identify the most stable EC miRNA. Then, the RefFinder [21] merging tool was utilized to compare and merge the output of the four algorithms.

### 2.7. Statistics Analyses

The statistical tests utilized are described in the figure legends and methods. The graphs and statistical tests were executed employing GraphPad Prism 8 (GraphPad Software Inc., San Diego, CA, USA).

## 3. Results

### 3.1. Serum RNA Quality/Quantity

The mean RIN (RNA Integrity Number) was 8.9, and the mean 260/280 ratio was 1.9 (range 1.88–2.00), which indicates high RNA purity. The amount of extracted RNA ranged from 9 to 16 ng/µL and was corrected before RT [15].

### 3.2. Stable EC miRNA Profiling Based on NGS miRNome Outputs

The comparison between the T0 miRNome and the T1 miRNome allowed us to identify eight stable miRNAs based on the mean stability value of count per million reads (CPM average), T0 versus T1 significance value (*p*-value) *p* ˃ 0.05, fold change (FC) value ˂ 1.1, and Bonferroni test value ˃ 1. The additional information is reported in Appendix A—NGS data (Appendix A). The results reported in Table 3 indicate the best EC miRNAs were selected in the first step of the experimental framework.

### 3.3. Bioinformatics Results of EC Expression Stability and EC Final Ranking

According to MIQE guidelines, the correct way to normalize miRNA qPCR data is by using more stably expressed EC miRNAs. The most stable candidate EC miRNAs were evaluated using four mathematical approaches: comparative Delta-Ct, BestKeeper, NormFinder, and GeNorm. As a result of each of the four bioinformatics methods, an EC miRNA (at T0 and T1) stability value (SV) is obtained (Figure 2). Additional information on the comparative ΔCt method, BestKeeper, Normfinder, GeNorm, and RefFinder are reported in the Appendix A. EC: expression stability.

Finally, RefFinder, based on the SV results of Delta-Ct, BestKeeper, NormFinder, and GeNorm, assigns an appropriate weight to the individual EC miRNAs by calculating the geometric mean of their weights for the overall T0 and T1 final ranking. RefFinder rank identified the top five miRNAs (gray box) at T0: miR-16, miR-191, miR-320, miR-93, and let-7b, while in T1, rank identified: miR-320, miR-191, let-7b, miR-16, and miR-93. MiR-191 and miR-320 (written in white in the dark gray box) maintain the best stability performances at both T0 and T1 (Table 4).

### 3.4. QPCR Data Quality Control and qPCR Data Normalization

To achieve an accurate identification of miRNA expression level, as a first hypothesis, the sources of technical variability were reduced by controlling the sample quality by inspecting the UniSp2, UniSp4, and UniSp6 amplification curves to remove outliers (Cq > 35). All tests did not show Cq values > 35. As the second step, following MIQE guidelines, qPCR data normalization with the most stable endogenous control miRNA should be performed to decrease the variability of experiments related to different amounts of starting RNA. The RefFinder algorithm ranked as the best EC normalizer: miR-16, miR-191, miR-320, miR-93, and let-7b (Table 4).

## 4. Discussion

This research evaluated different normalization strategies used for quantifying circulating miRNAs in the serum of working dogs trained to search for lost people after a natural catastrophe. As far as the GdF-trained dogs are concerned, SAR simulation represents a state of stress that dogs are able to regain physiologically as a result of training and exercise [15]. The SAR-learning-course includes acquisitions related to the enriched living environment in which dogs improve their problem-solving abilities, improve their physical performance, and develop better relationships by listening to and obeying the handler. In dogs, as in humans, physical activity triggers epigenetic changes in pathways associated with energy metabolism, insulin sensitivity, and more, regulating body homeostasis during exercise [22]. This adaptive physiological response is based on epigenetic regulation of gene expression mediated by miRNAs [23]. Changes in circulating miRNA levels in response to exercise allow adaptive regulation of the physiological processes including regulation of contraction and calcium signaling [24,25], bone metabolism [26,27], myocardial, skeletal muscle metabolism, repair and remodeling, mitochondrial physiology, inflammation, and angiogenesis [28]. Taking into consideration the importance of these epigenetic studies on the expression level of circulating miRNAs, a rigorous definition of the most stable normalizer is needed to reduce the number of confounding factors that influence the analytical outcome of the qPCR data.

To give relevance to the post-analytical normalization-dependent variability, this study paid special attention to stable circulating miRNA characterization by NGS profiling in a cohort of SAR dogs during two physiological states: resting and after simulated SAR performance. The NGS output data allowed simultaneous exploration and subsequent comparison of circulating miRNAs at time points T0 and T1. The stringent selection to improve the NGS data quality allowed us to identify a subset of cmiRNAs with stable expression in the T0 and T1 miRNomes that could be considered probable ECs. The selected cmiRNAs that were not differentially expressed were then validated by qPCR in a larger cohort of SAR dogs. The NGS-selected cmiRNA panel comprising cfa-miR-486, cfa-miR-16, cfa-miR-92, cfa-miR-191, cfa-miR-223, cfa-miR-423, cfa-let-7b, cfa-miR-25, and cfa-miR-93 was enriched with cfa-miR-320 because the previous studies showed that cfa-miR-320 is remarkably stable in serum [15]. In a panel of 179 circulating miRNAs, Falardi et al. [29] identified hsa-miR-320d (which has 100% homology with cfa-miR-320) as the most appropriate EC reference for circulating miRNA qPCR data normalization. The research of Falardi et al. emphasizes the stability of hsa-miR-320d in the basic physiological and physically active states due to the low association between miR-320 and the circulating miRNAs involved in exercise remodeling. The relevance of the claim is supported by the finding that the physical activity state, as opposed to the basic physiological state, undergoes epigenetic remodeling of skeletal muscle and bone [29]. The let-7 miRNA family, defined as one of the most conserved miRNA families in different species [30], is frequently deregulated in a wide variety of diseases in the dog as well as in other mammals [31]. The elevated presence of let-7b is considered strongly predictive of canine myxomatous mitral disease [32] or the poor outcome following ischemic stroke [33], and this probably supports the fact that let-7b in physiologically healthy dogs is a stable and reliable EC.

Research over two decades has demonstrated that qPCR data need to be normalized against an unregulated endogenous reference control gene to minimize variation that can mask or exaggerate biologically significant differences in gene expression levels. Extensive research on gene expression shows that using endogenous reference genes selected with bioinformatics criteria prior to conducting gene expression studies increases the accuracy of relative quantification. Conscious of this, to ensure the accurate selection of reference ECs, the putative ECs selected in the T0 and T1 miRNome were validated by qPCR in an extended sample cohort. Then the most acclaimed bioinformatics approaches, such as the Delta-Ct method, BestKeeper, NormFinder, and GeNorm, were used to choose the best EC reference gene [34,35]. In the final analysis, the RefFinder algorithm, based on the rankings of Delta CT, BestKeeper, NormFinder, and GeNorm, assigned an appropriate weight to the individual miRNAs by calculating the geometric mean of their weights and getting the T0 final EC ranking: miR-16, miR-191, miR-320, miR-93, and let-7b; and the T1 final EC ranking: miR-320, miR-191, let-7b, miR-16, and miR-93. Taking into consideration the aim of this study, to prevent an experimental bias that invalidates output data during extraction before reverse transcription and qPCR amplification, the reference exogenous miRNAs spike-in were added to control RNA quality and RT-qPCR reaction efficiency.

The optimal EC should be unrelated to biological variations, stages of disease, or treatments, and have storage stability, extraction properties, and quantification efficiency similar to the target miRNA. Therefore, before carrying out an experiment, it is imperative to first determine the best EC, keeping in mind that the EC should be expressed approximately at the same level as the target gene. Although many reports describe the ideal EC miRNAs in different species, tissues, or diseases, it is to be pointed out that, to date, there has been a lack of research describing the selection of circulating EC miRNAs in working dog serum.

A number of normalisation approaches have been suggested, but the use of one EC, selected as the best by Delta-Ct, BestKeeper, NormFinder, and GeNorm bioinformatic methods, is currently the favorite approach for the selection of the best EC to normalize qPCR data. Although some authors prefer to normalize with the average Cq value (called the mean centering method) derived from the means of different endogenous ECs [36]. In our opinion, normalization by the mean centering method should be limited to studies that analyze a large number of miRNAs. A variant of this method is the restricted mean centering method, developed for experiments in which a substantial proportion of miRNA data values are lacking. This method minimizes the technical variance in normalized expression values by utilizing a small number of normalizing ECs [20,37,38]. Other authors also suggest normalization with exogenous synthetic miRNAs added during the RNA extraction process. In our opinion, normalization with exogenous synthetic miRNAs (such as spike-ins) only correct variables linked to the extraction method, RT, or qPCR, but not for intrinsic analytical variables related to the initial sample quantity, collection method, and storage conditions. At present, a universally accepted normalization strategy is still lacking, and, unfortunately, the diverse approaches in use provide different results, with a high risk of confounding.

The differing results for the selection of the best EC, derived from Delta-Ct, BestKeeper, NormFinder, and GeNorm bioinformatic approaches show that each method has its strengths and weaknesses, so a good strategy might be to let all of them contribute to the best EC identification [27]. We are confident that this study not only provides a helpful result in itself but also gives an experimental design for selecting the best endogenous controls to normalize gene expression, not only in circulating miRNAs.

## 5. Conclusions

Detecting gold-standard EC is crucial for producing accurate qPCR results. Our study showed the criteria for identifying appropriate endogenous controls to normalize the qPCR expression data of circulating miRNAs. The research was performed on search and rescue dogs under resting conditions and in response to SAR physiological stress. Possible circulating miRNAs as endogenous controls were selected through NGS followed by qPCR validation. The best-known algorithms Delta Ct, BestKeeper, NormFinder, GeNorm, and RefFinder were used to rank cmiRNAs in order of stability. miR-191 and miR-320 appeared to be the most stable serum endogenous controls. We are confident that this study provides a helpful result on the best-circulating EC miRNAs and provides agile and reliable criteria for identifying suitable endogenous controls to normalize qPCR expression data.

## Figures and Tables

**Figure 1 animals-13-00576-f001:**
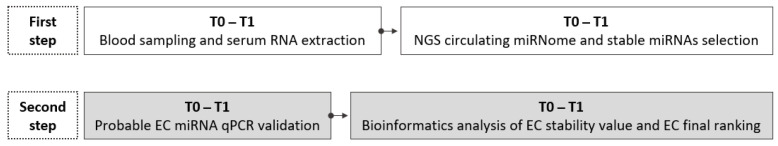
Research flowchart. In the first step, in a sample size of six dogs, a subset of miRNAs is identified by NGS as stable miRNAs. The white box shows all the phases performed in the first step: blood sampling at T0 and T1, serum RNA extraction, NGS circulating miRNA profiling, and stable miRNA selection. In the second step (gray boxes), the phases are stable miRNA qPCR validation (22 dogs) and bioinformatics analysis of qPCR output data to determine EC miRNAs’ stability ranking.

**Figure 2 animals-13-00576-f002:**
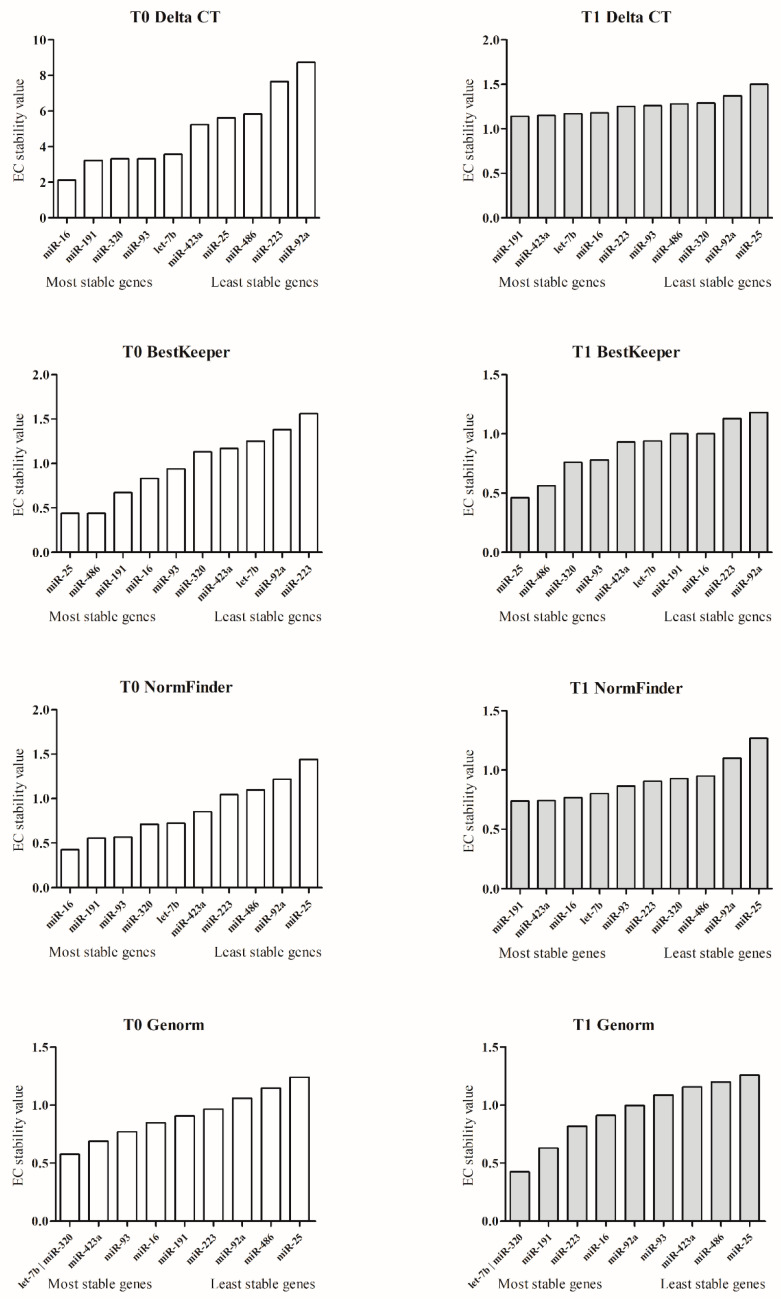
MiRNA stability value. The figure shows the EC miRNA stability value resulting from the Delta-Ct, BestKeeper, NormFinder, and GeNorm mathematical approaches. The left column plots (white) show the stability of miRNAs evaluated at T0, and the right column plots (grey) the miRNA stability assessed at T1. The miRNAs on the *x*-axis, near the origin, are the most stable; conversely, the miRNAs that are distant are the least stable.

**Table 1 animals-13-00576-t001:** SAR dog characteristics. The table shows the distribution and characteristics of the SAR dogs (n = 22); dog identification (ID) has its letter, and the gender is indicated in brackets (M)/(F).

Dog ID	Age	Weight	Breed
A (F)	3	25.4	Belgian Shepherd
B (F)	2	22.7	Belgian Shepherd
C (F)	2	26.1	Belgian Shepherd
D (M)	2	27.3	Belgian Shepherd
E (M)	3	30.3	Belgian Shepherd
F (M)	3	18.3	Border Collie
G (F)	2	21.4	Belgian Shepherd
H (F)	2	25.4	Belgian Shepherd
I (M)	3	28.3	Belgian Shepherd
j (M)	3	27.6	Belgian Shepherd
k (F)	3	15.4	Border Collie
L (M)	3	27.2	Belgian Shepherd
M (M)	2	29.2	Belgian Shepherd
N (F)	3	27.1	Belgian Shepherd
O (M)	3	26.8	Belgian Shepherd
P (F)	2	25.7	Belgian Shepherd
Q (M)	2	26.8	Belgian Shepherd
R (M)	2	16.3	Border Collie
S (F)	2	27.1	Belgian Shepherd
T (M)	3	17.2	Border Collie
U (F)	3	27.6	Belgian Shepherd
V (F)	3	28.1	Belgian Shepherd

**Table 2 animals-13-00576-t002:** QPCR-specific pair primers. The resuspended primer mix contains both forward and reverse sequences. * Previous studies have identified miR-320 as the best EC miRNA in SAR dog serum.

Primer	Recommended for:	GeneGlobe ID
UniSp2	Spike-in (RNA isolation efficiency evaluation)	YP00203950
UniSp4	Spike-in (RNA Isolation efficiency evaluation)	YP00203953
UniSp6	Spike-in (RT and PCR inhibitors evaluation)	YP00203954
miR-320 *	Serum EC miRNAs	YP00206042
miR-486	Probable serum EC miRNAs	YP02119777
miR-16	Probable serum EC miRNAs	YP00205702
miR-92a	Probable serum EC miRNAs	YP00204258
miR-191	Probable serum EC miRNAs	YP00205972
miR-223	Probable serum EC miRNAs	YP00205120
miR-423a	Probable serum EC miRNAs	YP00205624
let-7b	Probable serum EC miRNAs	YP00204750
miR-25	Probable serum EC miRNAs	YP00204361
miR-93	Probable serum EC miRNAs	YP00204715

**Table 3 animals-13-00576-t003:** Stable EC miRNA. The table reports the NGS results of the dog serum miRNome evaluated at rest (T0) and after physiological stress conditions due to a SAR performance simulation (T1). T0 CPM and T1 CPM columns show the mean values of the miRNAs selected as probable ECs. The T0 vs T1 column shows the values of Max group means (MGM), fold change (FC), Log2 fold change, *p*-value, FDR *p*-value, and the results of the Bonferroni test.

MiRNA	T0	T1	T0 vs T1					
	CPM Average	CPM Average	MGM	FC	Log_2_ FC	*p*-Value	FDR *p*-Value	Bonferroni
cfa-miR-486	284,055	344,565	344,565	1.209	0.274	0.251	0.979	1.000
cfa-miR-16	33,942	33,948	33,948	−1.019	−0.027	0.863	0.995	1.000
cfa-miR-92a	22,861	26,027	26,027	1.132	0.179	0.333	0.979	1.000
cfa-miR-191	18,829	17,113	18,829	−1.166	−0.221	0.154	0.979	1.000
cfa-miR-223	12,581	12,293	12,581	−1.106	−0.145	0.331	0.979	1.000
cfa-miR-423a	11,778	13,313	13,313	1.168	0.225	0.110	0.979	1.000
cfa-let-7b	9731	11,258	11,258	1.111	0.151	0.221	0.979	1.000
cfa-miR-25	8826	10,131	10,131	1.100	0.137	0.239	0.979	1.000
cfa-miR-93	5698	6336	6336	1.054	0.076	0.491	0.979	1.000
cfa-miR-320	1040	1098	1098	−1.013	−0.019	0.929	0.995	1.000

**Table 4 animals-13-00576-t004:** RefFinder comprehensive ranking. The tables show the ranking of stability values assigned by RefFinder at T0 (up) and T1 (down). The RefFinder method recommends a comprehensive ranking that rates miRNAs as best, good, and average by ordering miRNAs from most stable to least stable. The first five miRNAs (gray box) have the best stability characteristics at both T0 and T1; in the dark gray box are the miRNAs with the best characteristics at both T0 and T1.

T0 RefFinder Ranking Order										
	Better			Good		Average				
**Delta CT**	miR-16	miR-93	miR-191	let-7b	miR-320	miR-423a	miR-223	miR-486	miR-92a	miR-25
**BestKeepeer**	miR-25	miR-486	miR-191	miR-16	miR-93	miR-320	miR-423a	let-7b	miR-92a	miR-223
**NormFinder**	miR-16	miR-191	miR-93	miR-320	let-7b	miR-423a	miR-223	miR-486	miR-92a	miR-25
**GeNorm**	let-7b	miR-320	miR-423a	miR-93	miR-16	miR-191	miR-223	miR-92a	miR-486	miR-25
**RefFinder**	miR-16	miR-191	miR-320	miR-93	let-7b	miR-423a	miR-25	miR-486	miR-223	miR-92a
**T1 RefFinder Ranking Order**										
	**Better**			**Good**		**Average**				
**Delta CT**	miR-191	miR-320	let-7b	miR-16	miR-223	miR-93	miR-486	miR-423a	miR-92a	miR-25
**BestKeepeer**	miR-25	miR-486	miR-423a	miR-93	miR-320	let-7b	miR-191	miR-16	miR-223	miR-92a
**NormFinder**	miR-191	miR-320	miR-16	let-7b	miR-93	miR-223	miR-423a	miR-486	miR-92a	miR-25
**GeNorm**	let-7b	miR-320	miR-191	miR-223	miR-16	miR-92a	miR-93	miR-423a	miR-486	miR-25
**RefFinder**	miR-320	miR-191	let-7b	miR-16	miR-93	miR-25	miR-486	miR-223	miR-92a	miR-423a

## Data Availability

Not applicable.

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
