# Peer review of "Identification of Appropriate Endogenous Controls for Circulating miRNA Quantification in Working Dogs under Physiological Stress Conditions"

_animals, 2023, doi:10.3390/ani13040576_

Round 1

Reviewer 1 Report

In this study, the author conducted an investigation project which aimed to select the appropriate circulating miRNAs as an endogenous control. The study model was the working dogs used to search and rescue missing persons after natural disasters.

This work addresses an important theme: the normalization of miRNA qPCR data. Therefore, the author’s objectives are 1) to provide suitable circulating EC miRNAs to study working dogs under resting conditions and SAR physiological stress and 2) to provide an appropriate method to select the most stable ECs to normalize qPCR data for future gene expression studies.

The introduction focus on the explanation of the miRNAs’ biology and stability. It is recommended to provide additional references in the first paragraph

The miRs that were considered the best EC normalizers included miR-16, miR-270 191, miR-320, miR-93 and let-7b. It is recommended that should be discussed that, for example, let-7b was found to be upregulated in canine myxomatous mitral disease as stated in:

Reis-Ferreira A, Neto-Mendes J, Brás-Silva C, Lobo L, Fontes-Sousa AP. Emerging Roles of Micrornas in Veterinary Cardiology. Vet Sci. 2022 Sep 28;9(10):533. doi: 10.3390/vetsci9100533. PMID: 36288146; PMCID: PMC9607079.

Author Response

Dear reviewer,

Thanks for the comments and suggestions. A revision of the manuscript has been carried out to take into account all your considerations. As suggested, we have included more appropriate references in the manuscript. The description of the method has been improved, and we have included the "Conclusions" paragraph in the revised manuscript. All changes have been made in red, please see the attached revised manuscript.

Best regards,

Gabriella Guelfi

Reviewer 2 Report

The authors present a work entitled "Identification of appropriate endogenous controls for circulat-2 ing miRNA quantification in working dogs under physiologi-3 cal stress conditions" in which the preliminary routine validation work of microRNAs is described for an investigation on the role of these molecules in search and rescue working dogs in response to physiological conditions of stress and rest. No data is presented regarding this model which is not mentioned in the summary in brief or in the abstract. Therefore, it remains exclusively a methodological proposal without any innovative element. Therefore, it remains a valid preliminary tool that can be used in a presentation of data on the role of putative circulating microRNAs in the regulation of stress in working dogs

Author Response

Dear Reviewer,

Thanks for the observation. As suggested, we have included, in the revised manuscript, more appropriate references, the description of the method has been improved, and we have included the conclusions paragraph. All changes have been made in red, please see the attached revised manuscript.

Best regards,

Gabriella Guelfi

Round 2

Reviewer 2 Report

The article reports moderate improvements compared to the first version that do not modify the judgment: the work carried out can be considered preliminary for a research on the role of circulating microRNAs under physiological stress conditions. There is minimal information about this and the standardization work, classic, could be done on any canine model.

Author Response

Dear reviewer, the sentence structure and grammar of the manuscript were revised as suggested. Please see the attached revised manuscript.
